# Anticoagulant Therapy in Patients with Antiphospholipid Syndrome

**DOI:** 10.3390/jcm11236984

**Published:** 2022-11-26

**Authors:** Marco Capecchi, Maria Abbattista, Alessandro Ciavarella, Mario Uhr, Cristina Novembrino, Ida Martinelli

**Affiliations:** 1Division of Hematology, Clinica Moncucco, 6900 Lugano, Switzerland; 2Department of Biomedical Sciences for Health, Università degli Studi di Milano, 20133 Milan, Italy; 3Angelo Bianchi Bonomi Hemophilia and Thrombosis Center, Fondazione IRCCS Ca’ Granda Ospedale Maggiore Policlinico, 20122 Milan, Italy; 4Department of Hematology, Synlab-Suisse, 6900 Lugano, Switzerland

**Keywords:** antiphospholipid syndrome, antiphospholipid antibodies, thrombosis, obstetrical complications, anticoagulation

## Abstract

Antiphospholipid syndrome (APS) is a systemic autoimmune disease characterized by the persistent positivity of antiphospholipid antibodies (aPLA) together with thrombosis or obstetrical complications. Despite their recognized predominant role, aPLA are not sufficient to induce the development of thrombosis and a second hit has been proposed to be necessary. The mainstay of treatment of APS is anticoagulant therapy. However, its optimal intensity in different presentations of the disease remains undefined. Moreover, decision on which patients with aPLA would benefit from an antithrombotic prophylaxis and its optimal intensity are challenging because of the lack of stratification tools for the risk of thrombosis. Finally, decision on the optimal type of anticoagulant drug is also complex because the central pathway responsible for the development of thrombosis is so far unknown and should be carried out on an individual basis after a careful evaluation of the clinical and laboratory features of the patient. This review addresses the epidemiology, physiopathology, diagnosis and management of thrombosis and obstetrical complications in APS, with a special focus on the role of direct oral anticoagulants.

## 1. Introduction

Antiphospholipid syndrome (APS) is a systemic autoimmune disease characterized by the persistent positivity of antiphospholipid antibodies (aPLA) together with thrombotic events (venous, arterial, or microvascular) or obstetrical complications.

The presence of aPLA, including lupus anticoagulant (LA), anti-cardiolipin (aCL) and anti-beta-2-glicoprotein I (aβ2GPI) IgG and IgM antibodies, represents a severe acquired thrombophilia abnormality comparable, for the magnitude of the increased risk of thrombosis, to such inherited factors as deficiencies of the anticoagulant proteins antithrombin, protein C or protein S and the gain-of-function homozygous G1691A mutation in factor V (factor V Leiden) or G20210A mutation in prothrombin gene. However, the risk of thrombosis varies according to the combination of positivity for aPLA. The isolated presence of aCL antibodies is associated with the lowest risk of thrombosis, while triple positive aPLA, defined as the concomitant positivity of LA, aCL and aβ2GPI antibodies independently of their isotype, is associated with the highest risk [1]. On the other hand, isolated LA positivity is the only single positivity associated with a high risk of thrombosis [2].

The first association between LA and aCL antibodies with thrombosis or miscarriages was firstly described in the early 1980s in patients with systemic lupus erythematosus (SLE) [3]. In the following years the association was also found in patients without SLE [4] and the first diagnostic criteria for APS were formulated [5].

The mainstay of treatment of APS is anticoagulant therapy. However, despite the increased awareness of APS management, the intensity of anticoagulation in different presentations of the disease and the usefulness of antiplatelet drugs are uncertain. Moreover, decisions on which patients with aPLA may benefit from an antithrombotic prophylaxis and its optimal intensity are challenging because of the lack of stratification tools for the risk of thrombosis. Finally, decisions on the optimal type of anticoagulant drug are also complex because the central pathway responsible for thrombosis development is so far unknown.

## 2. Epidemiology

The estimated annual incidence of APS is approximately 1–2 per 100,000 person-years with a mean age at diagnosis of 50 years [6,7], which raises rapidly after 60 years. A higher annual incidence rate has been reported in women than in men, particularly between 11 and 70 years [8]. The prevalence of persistent high titers of aPLA in the general population is 1–5% [9,10], but only a minority of individuals develops APS. The prevalence rises to 12–30% in patients with SLE and to 5–15% in unselected patients with VTE, with a 10 to 15-fold increased relative risk of venous and arterial thrombosis [11]. However, the risk estimation varies widely in different studies. LA is strongly associated with thrombosis, aβ2GPI antibodies show a modest association and aCL antibodies are not significantly associated. The concomitant positivity of two different antibodies is associated with a higher risk of thrombosis and in triple positivity the risk is as high as 5.3% per year [12], three-fold higher than in double positivity [13]. The risk of thrombosis in patients with isolated aPLA single positivity is low with an incidence of 1.3% patient-years [12]. Evidence on the different risks associated with aCL and aβ2GPI immunoglobulin isotype (IgG or IgM) and antibody titer is less robust, but more significant correlations with thrombosis have been found for the IgG than for the IgM isotype of aCL and aβ2GPI antibodies [14]. Overall, a low-risk aPLA profile is defined by isolated positivity of aCL or aβ2GPI antibodies, and a high-risk profile by isolated LA positivity, double or triple positivity.

The clinical variants of the syndrome include vascular APS, characterized by either venous or arterial thrombosis (approximately 80% of prevalence) and obstetrical APS, characterized by such complications as recurrent miscarriages, fetal death, preeclampsia, abruptio placentae or intrauterine growth restriction (approximately 20% of prevalence). Women mostly develop venous thrombosis and/or obstetrical complications at a young age, while men mainly suffer arterial thrombosis, often recurrent, later in life [8,15].

A rare variant, observed in less than 1% of cases of APS [16], is represented by the catastrophic antiphospholipid syndrome (CAPS), a life-threatening condition characterized by the development of systemic micro-thrombosis. It is typically triggered by a precipitating factor, particularly infections (45% of cases) [17] and can be fatal in up to 45% of patients if not treated promptly [18,19].

Antiphospholipid syndrome is a primary condition in approximately half of patients (primary APS), otherwise it occurs in association with autoimmune diseases (secondary APS), particularly SLE, which is reported in approximately 35% of cases of APS [16,20,21] and usually affects young women, whereas APS in the elderly is rarely associated with SLE [17].

Recurrent thrombosis is common in patients with APS, even despite antithrombotic prophylaxis (recurrence rate of 30–40% during long term follow-up) [22]. The strength of the association between recurrent thrombosis and aPLA depends on the number of positive tests. Triple positivity of aPLA is predictive of worse outcomes. In particular, the cumulative incidence at 10 years of follow up in triple positive patients has been estimated at 45% and 47% for venous and arterial thrombosis and at 37% for obstetrical complications [22]. Recently, in a cohort of 312 patients with primary APS the rate of recurrent thrombosis was 46% (59% were triple positive and 70% on antithrombotic prophylaxis) and that of bleeding episodes 8.6% [23].

## 3. Physiopathology of Thrombosis in APS

The occurrence of thrombosis in patients with APS is described by a theory called the ‘two-hit model’. Indeed, only a portion of patients with aPLA develops thrombosis, suggesting that a ‘second hit’ or ‘trigger’ is needed to push the hemostatic balance toward clot formation. These second hits include environmental (e.g., infection, surgery, immobilization), inflammatory (e.g., autoimmune diseases) or other non-immunological procoagulant factors (e.g., oral contraceptive use) [24]. A genetic predisposition, involving genes encoding for the human leukocyte antigen (HLA) locus, has also been theorized to contribute to the development of thrombosis in patients with APS [25].

The major target of aPLA antibodies is β2GPI, a circulating protein binding phospholipid surface. This bond induces the release of endothelial procoagulant microparticles and an increased expression of the soluble forms of E-selectin, intercellular and vascular cell adhesion molecule 1 (ICAM-1 and VCAM-1) and tissue factor mRNA [26]. Moreover, this bond increases thrombin generation [27] and induces an acquired activated protein C resistance [28].

Regulation of fibrinolysis represents another mechanism for thrombus formation in APS. In fact, annexin A2, a receptor for tissue plasminogen activator and plasminogen, mediates the pathogenic effects of aPLA [29].

Complement activation has also been implicated in the pathogenesis of APS. Although the mechanisms of complement activation in APS are still poorly understood, the regulation of complement by means of the enhancement of C3/C3b degradation [30] and the activation of the classical complement pathway [31] induced by aPLA have been proposed.

Antiphospholipid antibodies may also induce thrombosis by directly interacting with platelets, inducing platelet aggregation and increasing glycoprotein IIb/IIIa activation and P-selectin surface expression [32]. On the other hand, platelets play a key role in the interaction between aPLA and endothelial cells [33].

Neutrophils are also involved in thrombus formation in patients with APS. Studies showed that plasma levels of cell-free DNA and NETs were higher in patients with primary APS than in healthy controls [34,35,36]. Similar results are reported in pregnant women with APS [37]. In addition, patients with APS showed a lower degradation of NETs compared to healthy controls [38].

Finally, aPLA may also induce tissue factor expression in monocytes [39] and activate a tissue factor signaling pathway by inducing the dissociation of an inhibited tissue factor coagulation initiation complex on the monocytes’ cell surface, thereby liberating factor Xa for thrombin generation [32].

## 4. Diagnosis

The occurrence of single or recurrent unexplained venous or arterial thrombosis, especially in young patients, obstetrical complications or both should raise clinical suspicion for APS. In patients with systemic autoimmune diseases, especially SLE, aPLA should be systematically searched.

The diagnosis of APS requires the combination of at least one clinical (i.e., venous and/or arterial thrombosis and/or adverse pregnancy outcome) and one laboratory criterion (i.e., the presence of persistent laboratory evidence of aPLA) following the revised Sapporo classification criteria (known as the Sidney criteria) and the International Society on Thrombosis and Haemostasis (ISTH) guidelines summarized in Figure 1 [40]. Diagnosis of APS is inaccurate in case of a period shorter than 12 weeks between the first and the confirmation aPLA positive test or if more than 5 years separate the clinical manifestation and the aPLA positive test.

The differential diagnosis of APS should include other causes of venous and arterial thrombosis and pregnancy morbidity. However, patients might have APS and concomitant transient risk factors for thrombosis (e.g., immobilization, oral contraceptive use, cardiovascular risk factors).

Catastrophic APS requires early diagnosis and aggressive therapy, but its diagnosis is often difficult because of the great variability of thrombotic manifestations that involve small blood vessels in different organs leading to multiple organ failure, simultaneously or over a short period. The proposed diagnostic algorithm includes a history of APS and/or persistent aPLA, ≥3 new organ thrombosis occurred within a week, biopsy confirmation of micro-thrombosis, exclusion of other causes of multiple organ thrombosis or micro-thrombosis (i.e., disseminated intravascular coagulation, heparin-induced thrombocytopenia, thrombotic microangiopathy) [41].

Although not considered as diagnostic criteria so far, some non-criteria aPLA (including anti-phosphatidylserine/prothrombin, anti-phosphatidylserine, anti-phosphatidylethanolamine and anti-annexin V antibodies) have shown the capacity to increase the diagnostic accuracy of APS and may become important in the future [42].

## 5. Primary Antithrombotic Prophylaxis

Patients with aPLA have an increased risk not only of recurrent, but also of first thrombotic events, and primary prophylaxis is of pivotal importance. Given that the risk estimation varies widely in different studies, a risk prediction model would be useful to identify patients with aPLA who may benefit from primary antithrombotic prophylaxis. A couple of risk scores, such as the antiPhosphoLipid Score (aPL-S) and the Global AntiPhospholipid Syndrome Score (GAPSS), have been proposed and validated in independent series of patients [43,44], but their applicability is limited to patients with underlying autoimmune diseases.

According to the most recent guidelines, primary antithrombotic prophylaxis with low dose (75–100 mg/day) aspirin (LDA) in high-risk profile aPLA patients without history of thrombosis is recommended. A metanalysis of 460 patients showed that those on LDA had a two-fold risk reduction of first thrombotic events than those without [45]. However, the meta-analysis included mainly observational studies and the level of evidence for this recommendation is low. On the other hand, the use of low molecular weight heparin (LMWH) in high-risk situations such as postoperative periods, lower limb fracture, immobilization, hospitalization, pregnancy/puerperium or central venous catheter placement is widely accepted, as for all other severe thrombophilia abnormalities. In our view, when the evidence of the efficacy of antithrombotic prophylaxis is uncertain, performing a complete thrombophilia work-up may help in the decision making.

Due to the high risk of venous and arterial thrombosis associated with the use of combined oral contraceptives in young women, their use should be avoided in patients with triple positive aPLA. In selected cases, if oral contraceptives are strongly needed, a concomitant anticoagulant therapy should be considered.

Even though some of the non-criteria aPLA have been proposed as potential biomarkers to predict the risk of thrombosis in APS [42], no recommendation on antithrombotic prophylaxis can be made based on their presence so far.

Finally, some data support the use of hydroxychloroquine to reduce the risk of thrombosis in patients with SLE with or without aPLA [46], but further studies are needed to investigate its efficacy in primary prevention of thrombosis in patients with primary aPLA positivity.

In patients with a high-risk aPLA profile and no history of thrombosis or obstetrical complications, prophylaxis with LDA is suggested, although with low evidence. The use of LDA was not associated with any improvement in obstetrical outcomes in pregnant women with aPLA [47]. Pregnant women with a history of thrombotic APS should be treated with LDA associated with LMWH at therapeutic doses. Those on secondary antithrombotic prophylaxis with VKA should switch to heparin promptly, preferably within the 6th gestational week. It has been proposed that women with a history of obstetric APS can receive an antithrombotic prophylaxis with variable intensity according to the type of pregnancy complications. In case of ≥3 spontaneous miscarriages or ≥3 fetal loss, LDA and heparin at prophylactic doses are recommended; in case of delivery <34 weeks of gestation for eclampsia, severe pre-eclampsia, or placental insufficiency LDA or LDA with heparin at prophylactic dose is suggested. In women with recurrent obstetrical complications despite LDA and heparin at prophylactic doses, intensification to therapeutic heparin is suggested and, in selected cases, addition of hydroxychloroquine, steroids or intravenous immunoglobulin may be considered [48].

## 6. Treatment of Venous Thromboembolism

A summary of the recommendations for the management of antithrombotic prophylaxis and treatment in patients with APS is reported in Figure 2.

For patients with APS and VTE, current guidelines recommend treatment with unfractionated heparin or LMWH followed by long-term vitamin K antagonists (VKA) with a target INR of 2.0 to 3.0 [49]. However, there is no consensus on the best strategy in terms of intensity of anticoagulation in different presentations of the disease and the usefulness of antiplatelet drugs, both in venous and arterial thrombosis.

If venous thrombosis occurs in pregnancy, LMWH at therapeutic doses periodically adjusted by body weight should be used over VKA, which are contraindicated in pregnancy because of their embryotoxic effect. Therefore, according to guidelines, women with a history of thrombotic APS who remain pregnant on VKA should be switched to therapeutic heparin doses before the 6th gestational week until delivery [50]. However, given the lowest risk in the second and third trimester compared to the first, in selected patients with a strong indication to VKA such as those with mechanic heart valves, the use of VKA can be considered in the second trimester, after completion of the embryogenesis. A new switch to heparin should be carried out around the 26th gestational week in view of delivery. Safety of direct oral anticoagulants (DOAC) during pregnancy is unclear due to lack of data [51].

In our opinion, when using heparin, the optimal strategy is to adapt therapeutic doses throughout pregnancy to increasing body weight. However, in some cases, an up-titration of the heparin dose beyond a weight-based approach should be considered. Indeed, pregnancy is characterized by an increase in heparin-binding proteins (that reduce its availability), an increased glomerular filtration rate (that increases its kidney elimination), an increased volume of distribution (with its dilution), an increased degradation by placenta enzymes and increased levels of prothrombotic coagulation factors such as fibrinogen and factor VII (that contribute to create a condition of heparin resistance) [52]. In this specific scenario, monitoring of anti-factor Xa activity may be helpful to titer the optimal heparin dose with the aim of reaching the desired anticoagulant effect to prevent recurrent thrombosis and minimize the risk of bleeding.

## 7. Treatment of Arterial Thrombosis

Patients with APS and arterial thrombosis outside the cerebral circulation should receive long-term anticoagulant treatment with VKA at an INR target of 2.0 to 3.0, as for those with venous thrombosis [48]. For patients with stroke, different strategies have been proposed depending on the individual risk profile. Elderly patients with a low titer of aCL antibodies on a single test may be treated with LDA alone. Despite LDA showing a similar rate of stroke recurrence as compared to warfarin in patients with APS and thrombosis, the results of the AntiPhospholipid Antibodies and Stroke-APASS study are not generalizable because patients with low titer aCL antibodies were also randomized to receive one or the other drug, and the INR target was even lower than 2.0 [53]. Patients with a high risk aPLA profile should receive VKA at an INR range of 2.0 to 3.0 with or without LDA, or alternatively VKA at an INR range of 3.0 to 4.0 [48]. The first regimen is the most accepted, considering that only few patients with arterial thrombosis were included in the clinical trials that compared different VKA intensity regimens [54,55]. In any case, antithrombotic treatment intensification should be reserved for patients with additional cardiovascular risk factors, progression, or recurrent thrombosis while on VKA with an INR range of 2.0 to 3.0 and a low risk of bleeding.

Risk-stratification tools are needed to identify patients at increased risk who may benefit from more aggressive antithrombotic treatment.

## 8. Secondary Antithrombotic Prophylaxis

Antiphospholipid syndrome is also characterized by a high risk of recurrent thrombosis, despite an adequate antithrombotic treatment. Among patients with unprovoked VTE, those with LA positivity have a 40% increased risk of developing recurrent thrombosis after discontinuation of anticoagulant therapy compared to patients without [56]. The rate of recurrent thrombosis in patients who remain anticoagulated after a first unprovoked VTE is 22.4% during the first 5 years and 23.5% during the following 5 years, with a 23.3% and 42.5% of patients maintained at a target INR > 3.0 [21]. As previously stated, several intensified anticoagulation regimens have been proposed, but there is no high-quality evidence to support one in particular except for switching to heparin during the acute phase [48]. For long-term anticoagulation, LDA addition or an increased INR therapeutic range are the available options. In particular, high-intensity regimens with an INR range of 2.5 to 3.5 or 3.0 to 4.0 have been proposed, but two randomized controlled trials showed no reduction of recurrent thrombosis [54,55]. In selected patients with contraindications to VKA at high INR range or to a combination of anticoagulant and antiplatelet therapy because of an increased risk of bleeding, as well as in patients with unstable INR, the use of the anti-factor II DOAC dabigatran can be considered. A recent meta-analysis showed comparable rates of recurrent venous and arterial thrombosis with dabigatran and VKA. However, the results should be interpreted with caution because only observational retrospective studies and case series were included [57].

In clinical practice, INR should be checked at the time of diagnosis of recurrent thrombosis, because a large proportion of patients recur at subtherapeutic INR levels. In these cases, a closer monitoring of INR can be an option. The possibility should also be considered that aPLA, especially LA, other than activated partial thromboplastin time, may also falsely prolong prothrombin time and INR, masking subtherapeutic VKA doses. This possibility is more common with point-of-care devices (10% of cases) rather than automated coagulometers because of the different thromboplastins employed for the test [58]. Some strategies to overcome this technical issue, such as measuring prothrombin levels or chromogenic factor X activity, have been suggested [59,60], but require a laboratory with great skills in coagulation tests.

In general, patients with APS have indication to continue lifelong anticoagulant treatment because of the high risk of recurrent thrombosis. This is valid in particular for patients with aPLA who developed an unprovoked event in the absence of transient and removable risk factors [56]. On the other hand, patients with APS who develop thrombosis in the presence of a transient risk factor no longer present may discontinue the anticoagulant therapy after at least three months, particularly if a complete recanalization is objectively documented. Anticoagulant therapy can also be discontinued in patients with a persistent aPLA negativization over time, even though evidence to support such an approach is lacking. Hence, it is very important to continue aPLA monitoring over time in patients on long-term anticoagulation. The decision to discontinue anticoagulant therapy in patients with APS is always challenging and should be taken on an individual basis and in accordance with patient’s preference.

More accurate risk-stratification models to identify patients at increased risk of recurrent thrombosis who may benefit from an intensification of the antithrombotic therapy are needed, always considering the consequent high risk of bleeding.

## 9. Direct Oral Anticoagulants

Direct oral anticoagulants have become the first line anticoagulant therapy for patients with VTE, due to their well-documented higher efficacy and safety and less food and drug interactions compared to VKA. In addition, the lack of monitoring is particularly useful in patients with LA due to its possible interference on INR test and highly appreciated by patients who are candidates for lifelong treatment. However, despite their proved efficacy for the treatment of VTE and the prevention of stroke in atrial fibrillation, their efficacy in strong prothrombotic stimuli such as APS is less robust. The first case reports and case series published in the literature showed contrasting results on the efficacy of DOAC in patients with APS. The RAPS (Rivaroxaban in AntiPhospholipid Syndrome) study was the first randomized trial aiming to evaluate the efficacy of rivaroxaban compared to warfarin (INR range of 2.0 to 3.0) as secondary prophylaxis in 116 patients with a previous episode of VTE. No recurrent thrombosis nor bleeding were reported in the two groups, but the study was inadequately powered for these clinical endpoints and the follow-up period was limited to 6 months. The primary surrogate endpoint was a global coagulation test represented by thrombin generation, that resulted as higher in the rivaroxaban than in the VKA group, but did not meet the non-inferiority threshold [61].

The first prospective study reporting data on clinical outcomes included 28 patients (15 on VKA and 13 on rivaroxaban) with a mean follow-up of 22 months and showed an increased rate of recurrent thrombosis in the DOAC group (19.4 vs. 2.4 per 100 patient years with a hazard ratio of 7.5). Recurrent events were arterial in four patients and venous in one and all patients who recurred were triple positive for aPLA [62].

The phase 3 TRAPS (Trial of Rivaroxaban in AntiPhospholipid Syndrome) trial evaluated the efficacy of rivaroxaban compared to warfarin (INR range of 2.0 to 3.0) in 120 triple positive APS patients with thrombosis and was prematurely interrupted for an excessively high rate of recurrent thrombosis (all arterial) in the rivaroxaban group [63]. These results have been confirmed in the two-year follow-up update of the same study, that showed a 33.3% rate of recurrent thrombosis in the DOAC group compared to a 5.7% rate in the warfarin group [64]. As a consequence of this evidence, the European Medicines Agency (EMA) stated that DOAC are not recommended (although not fully contraindicated) in patients with APS, particularly triple positive, and this recommendation was subsequently endorsed by the Food and Drug Administration (FDA).

Another phase 3 trial comparing rivaroxaban to warfarin (INR range of 2.0 to 3.0) in 190 patients with thrombotic APS confirmed an increased rate of recurrent thrombosis in the rivaroxaban group (3.9% vs. 2.1%), with arterial events occurring only in patients taking rivaroxaban [65].

The ASTRO-APS (Apixaban for Secondary Prevention of Thromboembolism Among Patients With AntiPhospholipid Syndrome) study randomized 47 patients to receive warfarin (INR range of 2.0 to 3.0) and apixaban and was prematurely interrupted because of a significantly increased rate of recurrent thrombosis (especially stroke) in the investigational arm (6 vs. 0 events) [66].

In general, all these studies highlighted an increased risk of recurrent thrombosis in APS patients treated with DOAC, especially in those with previous arterial thrombosis, and particularly a higher risk of arterial recurrent events. It is not known if the standard DOAC doses are sufficient to treat or prevent arterial thrombosis and consequently if higher doses are needed. The lower efficacy of DOAC compared to VKA in the prevention of arterial thrombosis in patients with APS may have some explanations. Preclinical studies in animal models showed that a deeper inhibition of the Xa activity is necessary to prevent arterial as compared to venous thrombosis [67]. Clinical data on this specific issue are not available and the RISAPS (RIvaroxaban for Stroke patients with AntiPhospholipid Syndrome) study, designed to answer this question by evaluating rivaroxaban 15 mg bid vs. warfarin (INR range of 3.0 to 4.0) in patients with APS and stroke, is ongoing (NCT03684564).

Although DOAC in APS are consistently declared less effective than VKA, a post-hoc analysis of the RE-COVER, RE-COVER II, and RE-MEDY trials showed that the safety and efficacy of dabigatran is not reduced in patients with aPLA [68]. This observation can be explained by the possible stronger mechanism of action of dabigatran, the only available anti-factor IIa anticoagulant, on the final common pathway of the coagulation cascade, but no data from specifically designed clinical trials are available so far.

With these studies as a background, international scientific societies have provided slightly different guidelines. The 2019 ESC (European Society of Cardiology) guidelines contraindicated the use of DOAC in patients with APS, independently of the type of thrombosis (venous or arterial), the intensity of aPLA positivity and with no distinction between the type of DOAC [49]. The same recommendations have been proposed by the 2020 ASH (American Society of Hematology) and NICE (National Institute for Health and Care Excellence) guidelines [69,70]. The 2020 ISTH (International Society on Thrombosis and Haemostasis) guidelines suggested the possibility to using a DOAC in patients already on DOAC for a venous thrombotic event in whom a single or double aPLA positivity is found, but after a critical discussion with the patient [71]. Similarly, the 2019 EULAR (European Alliance of Associations for Rheumatology), the 2020 BSH (British Society of Hematology) guidelines and the 2020 recommendations of the 16th International Congress on Antiphospholipid Antibodies Task Force Report on Antiphospholipid Syndrome Treatment Trends contraindicated the use of DOAC in patients with APS and arterial thrombosis or triple aPLA positivity, taking into consideration their use in patients with venous thrombosis and single or double aPLA positivity [48,72].

Although available data suggest that DOAC exposure during pregnancy is not associated with a significant risk of embryopathy [73], their use is discouraged by all international guidelines. Hence, DOAC should not be considered to treat thrombosis in pregnant women with APS or to prevent obstetrical complications in women with aPLA.

In conclusion, current evidence and guidelines pronounce against the use of DOAC in APS patients with triple positivity or with arterial thrombosis. On the other hand, in patients with APS with VTE and a single or double aPLA positivity, DOAC can be considered on an individual basis. Despite that current evidence is insufficient to make recommendations, if the choice of the anticoagulant drug falls on a DOAC, dabigatran may be preferred over the other anti-factor Xa DOAC.

## 10. Conclusions

Patients with vascular APS should receive life-long anticoagulant therapy if aPLA are persistent, whereas asymptomatic patients with aPLA should benefit from an antithrombotic prophylaxis in high-risk situations. Pregnant women with vascular APS on oral anticoagulant therapy must switch promptly to therapeutic heparin doses, possibly within the 6th gestational week. Pregnant women with obstetrical APS should receive LDA, prophylactic heparin doses or both during the whole gestational period. Decision on the intensity and duration of antithrombotic prophylaxis in high-risk situations is challenging because of the lack of risk-stratification tools. The choice of the most appropriate type of anticoagulant should be made on an individual basis after a careful evaluation of the clinical characteristics and the laboratory features of the patient.

## Figures and Tables

**Figure 1 jcm-11-06984-f001:**
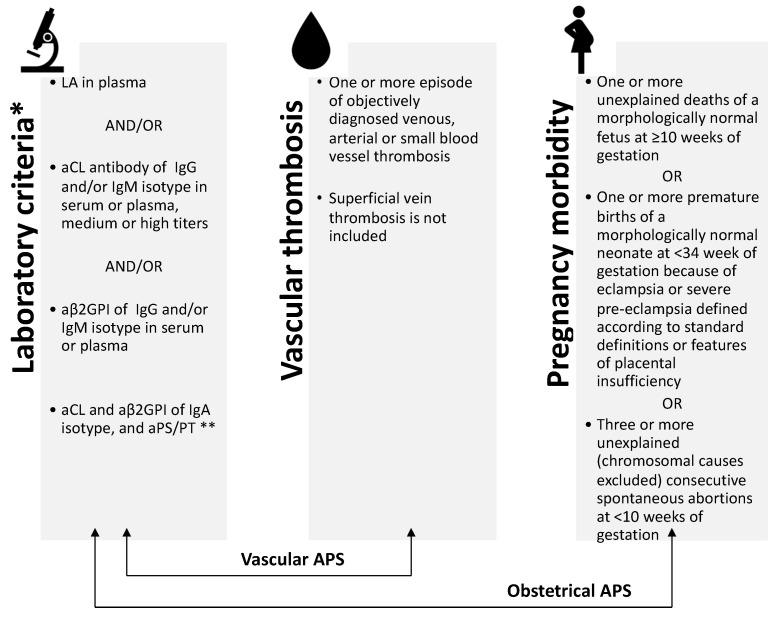
Classification criteria of antiphospholipid syndrome. LA, lupus anticoagulant; aCL anti-cardiolipin antibodies; aβ2GPI, anti-beta-2-glicoprotein I antibodies; aPS/PT, phosphatidylserine/prothrombin complex antibodies; APS, antiphospholipid syndrome. * detected on two or more occasions at least 12 weeks apart. ** clinical relevance is currently debated.

**Figure 2 jcm-11-06984-f002:**
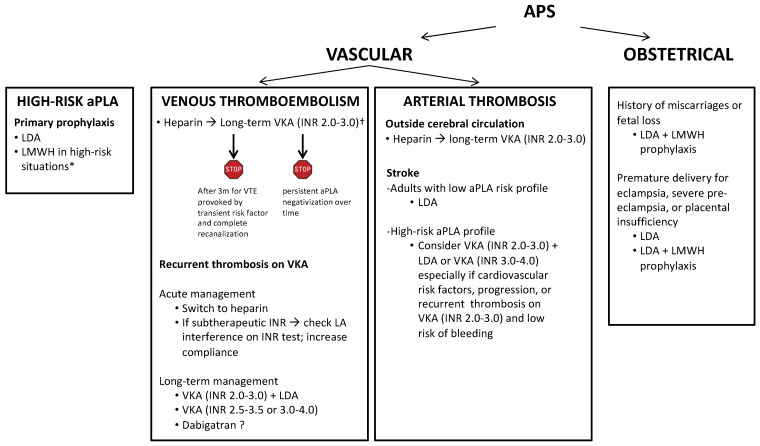
Recommendations for the management of antithrombotic prophylaxis and treatment in antiphospholipid syndrome. APS, antiphospholipid syndrome; aPLA, antiphospholipid antibodies; LDA, low dose aspirin; LMWH, low molecular weight heparin; VTE, venous thromboembolism; VKA, vitamin K antagonist; INR international normalized ratio; LA, lupus anticoagulant; CV cardiovascular. APS, antiphospholipid syndrome; aPLA, antiphospholipid antibodies; LDA, low dose aspirin; LMWH, low molecular weight heparin; VTE, venous thromboembolism; VKA, vitamin K antagonist; INR international normalized ratio; LA, lupus anticoagulant. * postoperative periods, lower limb fracture, immobilization, hospitalization, central venous catheter, pregnancy/puerperium. † DOAC (dabigatran preferred) can be considered in patients with single or double aPLA positivity (2019 EULAR; 2020 BSH; 16th International Congress on Antiphospholipid Antibodies Task Force Report on Antiphospholipid Syndrome Treatment Trends).

## Data Availability

Not applicable.

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
