# Peer review of "Anticoagulant Therapy in Patients with Antiphospholipid Syndrome"

_jcm, 2022, doi:10.3390/jcm11236984_

Round 1

Reviewer 1 Report

This is an exciting and brief summary of the current evidence available about anticoagulation in APS. It is preceded by extensive information about the syndrome. The author accomplishes the objective of the review by evaluating the anticoagulant options with some mentions of other treatment possibilities. 

These are my suggestions in order to improve the manuscript:

1.     At the XI ICAPA held in Sydney (Australia) in 2004, some small modifications were made to the Sapporo criteria that resulted in the current criteria in use since its publication in 2006: the Sydney criteria. I suggest addressing this point. These are the suggested references:

Miyakis, S. et al. International consensus statement on an update of the classification criteria for definite antiphospholipid syndrome (APS). J Thromb Haemost 4, 295-306 (2006).

Gomez-Puerta, J.A. & Cervera, R. Diagnosis and classification of the antiphospholipid syndrome. J Autoimmun 48-49, 20-25 (2014).

2.     The standard treatment for pregnancy complications is LMWH 0.4–06 mg/kg/day (“prophylactic” dose) since the positive pregnancy test combined with preconception daily LDA at least one month before starting attempts for a new pregnancy. The timing is relevant for good pregnancy outcomes. The manuscript may benefit from a more detailed treatment of the obstetric aPS. Suggested review: 

Alijotas-Reig J, Esteve-Valverde E, Anunciación-Llunell A, Marques-Soares J, Pardos-Gea J, Miró-Mur F. Pathogenesis, Diagnosis and Management of Obstetric Antiphospholipid Syndrome: A Comprehensive Review. J Clin Med. 2022 Jan 28;11(3):675.

I feel that the manuscript has some unresolved questions of the manuscript:

What is the value of non-criteria antiphospholipid antibodies considering primary thrombo-prophylaxis?

There are any general or non-pharmacological recommendations for these patients? For example, mobilization for long trips?

Minor recommendations: 

Figure 2 needs adjustments in the arrows, word size and elimination of free spaces. 

Reviewer 2 Report

In the manuscript entitled “Anticoagulant therapy in patients with antiphospholipid syndrome”, antiphospholipid syndrome (APS) was described in detail, from all aspects, including epidemiology, pathophysiology of thrombosis, diagnosis, primary and secondary antithrombotic prophylaxis, treatment of venous and arterial thrombosis, included consideration for oral anticoagulant therapy.

In order to improve this manuscript I would have a few specific comments:

1. This manuscript is too extensive and its size is more suitable for recommendations for the treatment of antiphospholipid syndrome, which is not the point of this text.

2. In the title, authors emphasized the use of anticoagulant therapy in patients with APS, so this should be followed in the manuscript itself.

3. The epidemiology of APS is not the focus of this manuscript. The authors should be based on the most important epidemiological data.

4. The pathophysiology of thrombosis in APS should also be presented in shortened form, especially the part related to complement. In this chapter, and also in other parts of the manuscript, do not start sentences with abbreviations such as in line 141 and line 146.

5. The diagnosis of APS is described in detail in the recommendations for the treatment of the APS and therefore this chapter should be shortened. In the Diagnosis section, the non-criteria manifestation can be omitted.

6. Primary and secondary prophylaxis of thrombosis, therapy of venous thromboembolism and arterial thrombosis are presented in detail with contemporary literature and have the important role as a modern therapy guideline.

 7. The role of direct oral anticoagulants (DOAC) in the primary and secondary prevention of thrombosis is very well presented.

8. The references are up to date, but I think they are too extensive. By shortening the parts of epidemiology, pathophysiology and diagnostics, the number of references would be more appropriate.

9. The conclusions correspond to the topic.
